# *AMPD1* C34T Polymorphism (rs17602729) Is Not Associated with Post-Exercise Changes of Body Weight, Body Composition, and Biochemical Parameters in Caucasian Females

**DOI:** 10.3390/genes11050558

**Published:** 2020-05-16

**Authors:** Agata Leońska-Duniec, Ewelina Maculewicz, Kinga Humińska-Lisowska, Agnieszka Maciejewska-Skrendo, Katarzyna Leźnicka, Paweł Cięszczyk, Marek Sawczuk, Grzegorz Trybek, Michal Wilk, Weronika Lepionka, Krzysztof Ficek

**Affiliations:** 1Faculty of Physical Education, Gdansk University of Physical Education and Sport, 80-336 Gdansk, Poland; leonska.duniec@gmail.com (A.L.-D.); kinga.huminska@gmail.com (K.H.-L.); maciejewska.us@wp.pl (A.M.-S.); k.leznicka@tlen.pl (K.L.); cieszczyk@poczta.onet.pl (P.C.); sawczuk_marek@wp.pl (M.S.); 2Department of Biomedical Sciences, Faculty of Physical Education, Jozef Pilsudski University of Physical Education in Warsaw, 00968 Warsaw, Poland; ewelina.jask@gmail.com; 3Department of Oral Surgery, Pomeranian Medical University, 72 Powstanców Wlkp. St., 70-111 Szczecin, Poland; g.trybek@gmail.com; 4Institute of Sport Sciences, The Jerzy Kukuczka Academy of Physical Education in Katowice, 40065 Katowice, Poland; 5EUAfME Drug Safety Unit (DSU) Regional Platform, Pfizer Poland, 02092 Warsaw, Poland; weronika.lepionka@gmail.com; 6Galen-Orthopaedics, 43-150 Bierun, Poland; krzysztof.ficek@galen.pl

**Keywords:** *AMPD1* gene, C34T, C34T polymorphism, sport genetics, physical activity, training, adaptation

## Abstract

Background: The C34T polymorphism (rs 17602729) in adenosine monophosphate deaminase 1 gene (*AMPD1*) is associated with muscular energy metabolism in exercise. However, the role of its potential modifying impact on exercise-induced changes in obesity related parameters is unknown. The aim of the study was to determine if the C34T polymorphism influences the effects of an exercise training. Methods: This study examines a group of one hundred and sixty-eight, young, non-obese Caucasian women in Poland who took part in a 12-week aerobic training program to determine the impact of allele and genotype distribution on training outcomes. Results: A two-way analysis of variance ANOVA was conducted assuming a dominant model by pooling rare homozygotes and heterozygotes (TT + CT, *n* = 79) and comparing against common homozygotes (CC, *n* = 89). Our results showed that the *AMPD1* C34T polymorphism was not related with selected parameters in study group. After completing the 12-week training program, a wide array of parameters (body mass, body mass index, fat mass, free fat mass, total body water) were significantly changed in the study participants with the exception of *AMPD1* genotypes, among whom no significant changes were observed. Conclusions: The results did not confirm that harboring the rs 17602729 T allele influences the effects of the training program.

## 1. Introduction

It is well known that the acute responses [1,2] as well adaptive changes in the human body following physical exercise show wide individual variance [3]. Consequently, changes in weight, body composition, and biochemical parameters in response to training programs may be more effective for some genotypes than others. However, the genetic background of these reactions still remains mostly unknown. Currently, some studies have tried to answer these questions, however, they still represent only the first steps towards an understanding of the gene variants x physical activity interactions [4,5,6]. An understanding of the genetic determinants is the most promising tool for individualizing training processes to improve competitive athletic performance, sport selection, sport traumatology, and even illegal “gene doping” [7].

Among the many regulators of energy metabolism in muscle, an integral part of the purine nucleotide cycle, adenosine monophosphate deaminase (AMPD, EC 3.5.4.6) is particularly important. The enzyme catalyzes the deamination of adenosine monophosphate (AMP) to inosine monophosphate (IMP) and shifts the reaction towards adenosine triphosphate (ATP) resynthesis in muscles [8].

Skeletal muscle-specific AMPD isoform M encoded by *AMPD1* gene located at 1p13.1, is stimulated after strenuous physical activity and is a key component of muscular energy metabolism while exercising [9,10]. A single nucleotide substitution of cytosine with thymine in nucleotide 34 of exon 2 (C34T, rs17602729), which results in the glutamine codon transforming into a stop codon, might cause the enzyme to become inactive [11]. Consequently, CC homozygotes have high skeletal muscle AMPD activity, CT heterozygotes have intermediate activity, and TT homozygotes have highly low activity (16% of the enzyme normal activity) [12,13]. Previous studies have reported that T allele is present in range 20–30% of Caucasians, however, only approximately 2% of the all Caucasian population is harboring TT genotype [11,12]. 

Since the data of Fishbein et al. [14], who proposed that an AMPD deficiency connected with T allele causes several exercise limitations, consequently it was described as the reason of metabolic myopathy, early muscle fatigue, stiffness, myalgia, and delayed recovery of muscular strength [9,10,14,15,16]. However, the T allele correlated with better outcomes among patients who had experienced heart failure (HF) and a greater rate of cardiovascular survival in patients with coronary artery disease (CAD) [17,18]. Additionally, Safranow et al. [19] described strong association of the mutated T allele with lower rates of obesity in patients with CAD and of hyperglycemia and diabetes in both CAD and HF patients. Therefore, it is possible that T allele reduces risk of obesity (abdominal), and in impact on the lower mass of visceral adipose tissue prevents insulin resistance and diabetes [19].

Previously, many authors described an association between this important genetic marker and athlete status. They suggested that the T allele is associated with a deficiency of AMPD and as a result, is effectively an exercise limitation [9,10,14,15,16,20,21,22]. Therefore, understanding the role of AMPD in muscle energy metabolism in physical activity is scientifically important. However, there is currently a lack of data concerning the potential impact of *AMPD1* on the extent and nature of the response to training in healthy women. Therefore, the main aim of the presented study was to determine whether the C34T polymorphism (rs17602729) in the *AMPD1* gene would influence the efficacy of an aerobic fitness program. To investigate the potential relationship between genotype and physical outcomes, we evaluated the allele and genotype distribution in young Caucasian females, whose selection criteria included specific body mass and composition characteristics, before and after engaging in a 12-week aerobic training program. Furthermore, changes in selected biochemical parameters were analyzed.

## 2. Materials and Methods

### 2.1. Ethics Statement

The Ethics Committee of the Regional Medical Chamber in Szczecin verified compliance of the investigation protocols (09/KB/IV/2011). The investigation protocols were conducted ethically according to World Medical Association Declaration of Helsinki and to Strengthening the Reporting of Genetic Association studies statement (STREGA). All participants were informed of the risks and benefits of the experiment protocols and gave written consent to genotyping. All personal information and results were anonymous. 

### 2.2. Participants

The research group included 168 Caucasian young females (age: 21 ± 1 years) of Polish nationality (body mass: 61 ± 2 kg, body height: 168 ± 2 cm). The inclusion criteria were as follows: (a) lack of engagement in regular physical activity during the early half year, (b) free from neuromuscular, metabolic and musculoskeletal disorders, (c) nonsmokers, (d) refrained from using medications and supplements. An individual diet program was assigned to each participant, which they were asked to follow (45−65% carbohydrates, 10−20% protein, 20−35% fat). Furthermore, participants were advised to maintain a daily cholesterol intake of less than 300 mg with a minimum dietary fiber intake of 25 g. The participants had three mandatory meetings where they received specific guidance on sensible nutrition and maintained their diet program throughout the entire experiment. The participants maintained a daily “diet diary”, recording everything they drank and ate during the program. Diet consultations were carried out every week, during which the quantity as well as the quality of the meals were assessed. As needed, minor changes were made [23].

### 2.3. The Measurements Body Composition 

The body mass and body composition measurements were made before and after the conclusion of a long-term (12 weeks) training program. The measurements were assessed using the bioimpedance method, which was performed using electronic scale Tanita TBF 300 M (Arlington Heights, IL, USA). During each measurement the following parameters were recorded [24]: Total body mass (BM; kg)Fat mass (FM; kg)Free fat mass (FFM; kg)Fat mass percentage (FM; %)Total body water (TBW; kg)Basal metabolic rate (BMR; kcal)Body mass index (BMI; kg/m^2^)

### 2.4. The Analyses of Blood Samples (Biochemical and Hematological) 

Blood sample analyses occurred before initiating the program of aerobic training and after its conclusion (36th training sessions). The samples of fasting blood were obtained from the antecubital vein in the morning and collected in 2 tubes. The blood count was analyzed using a 2.6 mL S-Monovette tube with K 3 EDTA (1.6 mg EDTA/mL blood) (SARSTEDT AG and Co.) The biochemical analyses were performed using a 4.9 mL S-Monovette tube consisting of ethylenediaminetetraacetic acid (K 3 EDTA; 1.6 mg EDTA/mL blood) and separating gel (SARSTEDT AG and Co., Nümbrecht, Germany). Whole blood was centrifuged 300× *g* for 15 min to obtain the blood plasma necessary for biochemical analysis. The analyses of the blood count were received using Sysmex K-4500 Hematology Analyzer (TOA SYSMEX, Kobe, Japan) and included: white blood cells (WBC); red blood cells (RBC); hemoglobin (HGB); hematocrit (HTC); mean corpuscular hemoglobin (MCH); mean corpuscular volume (MCV), mean corpuscular hemoglobin concentration (MCHC), and total platelet level (PLT). The biochemical analyses were conducted using Random Access Automatic Biochemical Analyzer (BIO-SYSTEMS S.A., Barcelona, Spain) for Clinical Chemistry and Turbidimetry A15. Blood plasma was used to analyze lipid profile: triglycerides (TGL); total cholesterol (TC); low-density (LDL) and high-density (HDL) lipoprotein concentrations. Plasma TGL and TC level were determined by diagnostic colorimetric enzymatic method (BioMaxima S.A., Lublin, Poland). The HDL plasma level was determined by human anti-ß-lipoprotein antibody and colorimetric enzymatic method (BioMaxima S.A.). All analysis procedures were verified by multiparameteric control serum (BIOLABO S.A.S, Maizy, France), and control serum of normal level (BioNormL) and high level (BioPathL) lipid profiles (BioMaxima S.A.) [24].

### 2.5. Training Program

Before the training program, each participant had their maximum heart rate (HRmax) assessed, on the basis of a continuous graded test on an electronic cycle ergometer (Oxycon Pro, Erich JAEGER GmbH, Hoechberg, Germany) [25]. All of the participants performed a continuous, graded exercise test on a cycle ergometer. The test began with 5 min of continuous cycling at a workload of 1.2 W/kg (frequency of 60 revolutions per minute RPM). After this phase, the workload was systematically increased by 15 watt [W] every minute until exhaustion. The trail ended when cycling frequency declined by 10%, that is, when the cycling frequency fell below 54 RPM. The heart rate measured at this point was noted as a HRmax [25]. Exercise intensity was carefully managed using, HR monitors to control each participant’s individual heart rate. The participants were instructed to maintain a HR or relative value of HRmax within designated ranges. The experimental sessions were preceded by three familiarization sessions, during which the exercise was performed for 30 min, at intensity of ~ 50% of HRmax. Each experimental training session consisted of a warm-up (10 min), aerobic exercise (43 min), and a cool-down phase (breathing-relaxing exercise with stretching; 7 min). Aerobic exercise was a combination of two styles including high and low impact. High impact styles include running, jumping and hopping, with a variety of flight phases. Low impact style is comprised of movements with at least 1 foot on the floor at all times. For both styles music of variable rhythm (tempo) intensity was used. A 12-week program of low-high impact aerobics exercise was divided as follows in Table 1 and all performed training sessions were supervised by a professional coach or instructor [26].

### 2.6. Genetic Analyses 

DNA was extracted from the buccal cells using a GenElute Mammalian Genomic DNA Miniprep Kit (Sigma, Darmstadt, Germany). Genotyping of the *AMPD1* rs17602729 polymorphism was performed using an allelic discrimination assay on a C 1000 Touch Thermal Cycler (Bio-Rad, Feldkirchen, Germany) instrument with TaqMan^®^ probes. To detect both alleles, TaqMan^®^ Pre-Designed SNP Genotyping Assays (Applied Biosystems, ‎Waltham, MA, USA) were used (assay ID: C__33603912_10), containing primers and fluorescently labelled (FAM and VIC) minor groove binder (MGB) probes.

### 2.7. Statistical Analyses

Allele frequencies were determined by means of gene counting. The chi-square statistical test was employed to test the Hardy-Weinberg equilibrium. The 2 × 2 (genotype × training interactions) repeated measures ANOVA was used to estimate the influence of the *AMPD1* rs17602729 polymorphism on aerobic training response. Data normality was verified with the Kolmogorov-Smirnov test. The level of statistical significance was set at *p* < 0.05.

## 3. Results

Genotype frequencies of the *AMPD1* rs17602729 polymorphism (CC – 53%, CT – 43%, TT – 4%) did not differ from Hardy-Weinberg expectations (Chi-square = 3.76, *p* = 0.052). A 2-way ANOVA was conducted assuming a dominant model by pooling rare homozygotes and heterozygotes (TT + CT, *n* = 79) and comparing against common homozygotes (CC, *n* = 89). Except for TC, TGL and LDL, all analyzed parameters were significantly altered in the course of the fitness program. Notably, the analyzed parameters did not differ according to the *AMPD1* genotypes. Furthermore, there were no genotype × training interactions (Table 2). 

## 4. Discussion

Our main findings are that the common C34T polymorphism (rs17602729) in *AMPD1* gene was not associated with selected body mass, body composition, and biochemical parameters in young, nonobese Caucasian women. Although, body mass, BMI, %FM, FM, FFM, TBW, HDL, and glucose changed significantly in the course of the 12-week aerobic training program, these parameters did not change across the *AMPD1* genotypes (genotype × training interaction). 

To the best of our knowledge, the presented study is novel in its analysis of the association between the *AMPD1* polymorphism and post-training body mass and composition, as well as biochemical parameter changes in physically active female. Therefore, the obtained results cannot be directly compared to other authors studies. However, when considering the effects of the analyzed *AMPD1* polymorphism on athlete performance, our previous studies conducted among Polish rowers, short-distance runners, short-distance swimmers, weightlifters and unrelated volunteers revealed a lower frequency of the *AMPD1* T allele in the group of athletes compared to controls, suggesting that harboring this allele is a negative factor to become an elite athlete [16,20]. These results confirmed many other studies which described that T allele is associated with a deficiency of AMPD and as a result exercise limitation [9,10,14,15,21,22]. Sabina et al. [9] related low skeletal muscle AMPD activity with a reduction in ATP and elevated adenosine concentration levels following exercise, as well as a prolonged rate of ATP resynthesis. A faster power decrease in individuals with AMPD deficiency was described during a Wingate cycling test [22]. Additionally, Rico-Sanz et al. [21] indicated that subjects with TT genotype have decreased exercise capacity and cardiorespiratory responses to cycling exercise in the sedentary state and in response to endurance training for 20 weeks. 

In contrast, Gross [27] revealed that a deficiency of detectable activity of AMPD in skeletal muscles did not show symptoms of myopathy. Additionally, Tarnopolsky et al. [28] described that participants with extremely low AMPD activity (<1%) demonstrated similar tolerance of the progressive cycle ergometry test to participants with normal AMPD activity. The studies undermined a role for the pathways thought to be altered by AMPD deficiency that could affect exercise performance [15]. The obtained results also did not show that the presence of a specific *AMPD1* genotype could be related with different post-exercise parametric changes.

Interestingly, the T allele, which is considered as a negative factor to become an elite athlete, also correlates with better recovery rates in patients with HF and CAD, as well as decreased frequency of obesity [17,18,19]. Safranow et al. [19] described lower value of: circumference of waist, waist to hip ratio, and BMI in a subject of patients harboring T allele with CAD, as well as hyperglycemia and diabetes in CAD and HF patients. Further investigation of the authors confirmed the association between the C34T polymorphism and lower prevalence of diabetes and obesity in patients with CAD or HF [29]. A proposed mechanism of the association between *AMPD1* genotype and obesity-related traits could be the impact of AMPD activity on the AMP-activated protein kinase (AMPK) activity, a key energy-sensing and signaling network that phosphorylates many proteins affected in cellular energy balance (e.g., by stimulating cellular glucose uptake) [30,31]. The major goal of AMP deaminase activity in skeletal muscles is reduction of AMP excess during physical activity leading to the AMP: ATP ratio decrease. Higher concentration of AMP, linked with its decrease deamination to IMP, could stimulate AMPK, and possibly prevent development of obesity and type 2 diabetes [32].

On the other hand, some studies did not confirm relationship between *AMPD1* genotype and obesity-related traits. The results obtained in HERITAGE Family Study indicated that there were no differences in BM or BMI across the genotypes [21], but this investigated group comprised 503 white, young, and healthy participants. Similarly, lack of significant association between the polymorphism and body mass, BMI, FM, FFM, or training index was recorded in 139 young, healthy, nonobese subjects [22]. In the study, differences in obesity-related traits across the *AMPD1* genotypes and genotype × training interactions were also not found. The possible explanation of differences in the results obtained from various studies is studied population-specific characteristics such as physical activity level, body weight, and diseases. Further investigations should be focused on participants who are overweight or obese to explain the role of the C34T polymorphism on development of obesity and diabetes, and to determine how effective the exercise program is.

The strengths of our study include precise experiment organization, consisting of careful regulation of both food intake and physical activity of a homogeneous Caucasian population, whose genotype distribution, body composition, and physiological and biochemistry parameters were analyzed at the outset and after the completion of a long-term (12-week) training program. A potential limitation of our study is the small size of the study group which might not show statistical power sufficient to yield meaningful analysis and interpretation. Additionally, low numbers of TT homozygotes, which is in accordance with other studies showed that ~2% of the general Caucasian population is homozygous (TT) [11,33,34], made us to analyze the TT homozygotes in combination with the CT heterozygotes, and thus may be potential confounders in the analyses. Another factor that should be considered as a limitation is population-specific characteristics such as relatively high PA levels as well as below-average body mass in the studied population. It needs to be highlighted that obesity is a polygenic trait. At present, over 700 genes and chromosomal regions have been specified as having influence in body weight and metabolic regulation [35]. The genetic marker assessed in isolation is unlikely to make a meaningful contribution to an “obesity phenotype” [4]. Additionally, the molecular mechanisms of change in response to physical activity in the human body is still unknown. Among the many factors influencing systemic response to exercise training are age, gender, diseases, individual predispositions, volume, intensity and frequency of physical activity, food intake, and others [4,36].

## 5. Conclusions

In conclusion, the present results are the first to reveal that weight, body composition, and biochemical parameters in young, nonobese Caucasian women do not change in response to a 12-week aerobic training program among carriers of the C34T polymorphism (rs17602729) in the *AMPD1* gene. Thus, the study did not confirm that harboring the rs17602729 T allele influences the effects of exercise training. The results of the presented study may not translate to other types of exercise with different intensity or training volume. However, further studies are necessary to elucidate the role of the C34T polymorphism on the effectiveness of an exercise training program, especially future analyses should be focused on participants with overweight and obesity.

## Figures and Tables

**Table 1 genes-11-00558-t001:** Experimental training program of low and high impact aerobics exercises.

Stage	Weeks	Training Sessions per Week (*n*)	Time of Each Training Session [min]	% of HRmax	Tempo - Beats Per Minute (BPM)
1	1–3	9	60	50%–60%	135–140
2	4–6	9	60	60%–70%	140–152
3	7–9	9	60	65%−75%	145–158
4	10–12	9	60	65%−80%	145–160

**Table 2 genes-11-00558-t002:** Training Adaptation with Respect to the *AMPD1* rs17602729 Polymorphism.

Parameter	CC (*n* = 89)		TT + CT (*n* = 79)		Genotype	Training	Genotype x Training Interaction
	Pre	Post	Pre	Post			
BM (kg)	60.6 ± 7.3	59.8 ± 7.5	60.6 ± 8.0	59.8 ± 7.9	0.959	<0.0001 *	0.983
BMI (kg/m^2^)	21.5 ± 2.1	21.3 ± 2.0	21.7 ± 2.8	21.4 ± 2.7	0.694	<0.0001 *	0.991
%FM (%)	23.7 ± 5.6	22.5 ± 5.0	24.1 ± 5.7	22.6 ± 6.3	0.784	<0.0001 *	0.308
FM (kg)	14.7 ± 4.8	13.8 ± 4.8	15.0 ± 5.3	14.0 ± 5.5	0.742	<0.0001 *	0.615
FFM (kg)	45.8 ± 3.1	46.2 ± 3.2	45.5 ± 3.4	46.0 ± 3.3	0.585	<0.0001 *	0.670
TBW (kg)	33.6 ± 2.7	33.9 ± 2.4	33.3 ± 2.5	33.7 ± 2.5	0.605	0.002*	0.668
TC (mg/dL)	170 ± 23	166 ± 26	170 ± 27	171 ± 28	0.575	0.379	0.178
TGL (mg/dL)	79 ± 26	81 ± 37	82 ± 38	87 ± 33	0.287	0.164	0.473
HDL (mg/dL)	63 ± 13	60 ± 13	67 ± 14	62 ± 14	0.153	<0.0001 *	0.221
LDL (mg/dL)	91 ± 22	90 ± 22	86 ± 22	91 ± 25	0.606	0.202	0.063
Glucose (mg/dL)	78 ± 10	75 ± 11	78 ± 10	76 ± 9	0.683	0.001 *	0.466

Mean ± standard deviation; * Statistically significant differences *p* < 0.05; BM = body mass; BMI = body mass index; BMR = basal metabolic rate; FM = fat mass; FFM = fat free mass; FM% = fat mass percentage; TBW = total body water; TC = total cholesterol; TGL = triglycerides; HDL = high-density lipoprotein; LDL = low-density lipoprotein.

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
