# Peer review of "AMPD1* C34T Polymorphism (rs17602729) Is Not Associated with Post-Exercise Changes of Body Weight, Body Composition, and Biochemical Parameters in Caucasian Females"

_genes, 2020, doi:10.3390/genes11050558_

Round 1
Reviewer 1 Report
An interesting paper in which the authors examine the effects of a training program on a C34T polymorphism. The results demonstrate that this polymorphism is not associated with training-induced changes in body composition. This is a well-written and interesting paper. I have several minor comments for the authors.
General comments:
Would it be wise to include C34T in the title? Furthermore, phrasing the title as a question seems to me to suggest that you would find an association between the polymorphism and the physiological parameters. As this wasn't the case, maybe the title needs rephrasing as a statement.
Why only use females as participants?
Line 124. Could you provide further details of the protocol for the continuous graded test.
Line 131. Why was this type of exercise training protocol used? Why not continuous aerobic training on a cycle ergometer, for example? Surely this would have been more controlled and easier to ensure everyone had the same training load? After reading the first line of this parapgraph, this is what I assumed the training would be.
Despite significant training induced changes, I would suggest that the training program wasn't actually particularly effective. Related to my point above, maybe a different training program would have seen more positive results and interaction with the genotyping.
Specific comments:
Lines 22-23. Should be "exercise-induced changes"?
Line 45. "Abbreviated" seems like an inappropriate word. It makes it sound like the exercise has been cut short.
Line 53. Should be "an AMPD deficiency".
Line 54. Should be "limitations".
Line 55. Delayed recovery of muscle what? Strength? Metabolism?
Line 109. "analysis".
Line 210. "similarly" would be a better choice of word than "alike".
Author Response
Reply– Thank you for your thoughtful and careful review of the manuscript. We believe that your suggestions and our revision will improve the quality of the manuscript. You will find your comments below followed by our point-by-point responses.
An interesting paper in which the authors examine the effects of a training program on a C34T polymorphism. The results demonstrate that this polymorphism is not associated with training-induced changes in body composition. This is a well-written and interesting paper. I have several minor comments for the authors.
General comments:
Would it be wise to include C34T in the title? Furthermore, phrasing the title as a question seems to me to suggest that you would find an association between the polymorphism and the physiological parameters. As this wasn't the case, maybe the title needs rephrasing as a statement.
Reply - The title was changed: “AMPD1 C34T polymorphism (rs17602729) is not associated with post-exercises changes of body weight, body composition, and biochemical parameters in Caucasian females”
Why only use females as participants?
Reply – Gender is an important factor differentiating the effectiveness of training programs, which is why we decided to examine only women. In the future we would like to replicate these results in a male cohort, but we are still trying to complete study group.
Line 124. Could you provide further details of the protocol for the continuous graded test.
Reply – the details of the protocol for the continuous graded test were added (L 143-148)
Line 131. Why was this type of exercise training protocol used? Why not continuous aerobic training on a cycle ergometer, for example? Surely this would have been more controlled and easier to ensure everyone had the same training load? After reading the first line of this parapgraph, this is what I assumed the training would be.
Reply – We fully agree with the reviewer that cycle ergometer can be more controlled. However this type of exercise is not a common form of training (exercise) performed by women. We decided to use aerobic training because it is the most common form of exercise for women. Therefore the results of the presented study are more similar to real world conditions.
Despite significant training induced changes, I would suggest that the training program wasn't actually particularly effective. Related to my point above, maybe a different training program would have seen more positive results and interaction with the genotyping.
Reply - Thank you for this comment. Yes, it is likely that another form of training may result in more favorable changes. We added such information to the conclusion (L 267-268)
Specific comments:
Lines 22-23. Should be "exercise-induced changes"?
Reply – the changes have been made
Line 45. "Abbreviated" seems like an inappropriate word. It makes it sound like the exercise has been cut short.
Reply – the changes have been made
Line 53. Should be "an AMPD deficiency".
Reply – the changes have been made
Line 54. Should be "limitations".
Reply – the changes have been made
Line 55. Delayed recovery of muscle what? Strength? Metabolism?
Reply – It should be "delayed recovery of muscle strength”
Line 109. "analysis".
Reply – the changes have been made
Line 210. "similarly" would be a better choice of word than "alike".
Reply – the changes have been made
Reviewer 2 Report
This study did an association analysis between C34T polymorphism in AMPD1 gene and the efficacy of an aerobic fitness program. The strength of this study is the large size of human samples. The weakness is that the experiment design is too simple to make a biological conclusion.
First, what is the necessity to do such an association analysis between AMPD1 polymorphism and aerobic fitness efficacy? It is mentioned in the introduction that AMPD1 polymorphism may be associated with obesity, insulin resistance and diabetes. In this scenario, the authors need to do an association analysis between AMPD1 polymorphism and BMI, GTT and ITT. It is not make sense to test aerobic fitness efficacy and recruit normal BMI people. The introduction needs to revise to introduce why the authors want to do such an analysis.
Second, it is not clear whether the diet is strictly controlled. The blood lipids and glucose are largely influenced by food intake. If the food is not strictly controlled, these data would be influenced by more than two factors (diet plus C34T and training).
Third, the conclusion that ‘AMPD1 C34T polymorphism is not associated with obesity-related traits’ is not solid. According to the introduction, it is expected that T allele reduces obesity. If normal BMI people were recruited in this study, treatments (such as HFD) should be utilized to induce the obesity of these people, and then check the obesity-related parameters. The combination of normal people and aerobic fitness did not explain anything about obesity, for the room is limited for normal people to lose weight or fatty acids.
Minors:
- In Abstract, the second sentence ‘However… is unclear’ is confusing. Please revise.
- In the training program, whether the cycle speed is the same for every participant?
- Consider to do a 3 × 2 [genotype (CC, CT and TT) × training interactions] analysis.
Author Response
Reviewer 2
This study did an association analysis between C34T polymorphism in AMPD1 gene and the efficacy of an aerobic fitness program. The strength of this study is the large size of human samples. The weakness is that the experiment design is too simple to make a biological conclusion.
Reply– Thank you for your thoughtful and careful review of the manuscript. We believe that your suggestions and our revision will improve the quality of the manuscript. You will find your comments below followed by our point-by-point responses.
First, what is the necessity to do such an association analysis between AMPD1 polymorphism and aerobic fitness efficacy? It is mentioned in the introduction that AMPD1 polymorphism may be associated with obesity, insulin resistance and diabetes. In this scenario, the authors need to do an association analysis between AMPD1 polymorphism and BMI, GTT and ITT. It is not make sense to test aerobic fitness efficacy and recruit normal BMI people. The introduction needs to revise to introduce why the authors want to do such an analysis.
Reply – Thank you for this comment. We were not specific enough and we added new senesces (L 40-48; 72-79)
It is well known that the adaptive changes in the human body in response to regular physical exercises show great individual variance. Consequently, changes in weight, body composition, and biochemical parameters in response to training programs may be more effective for some genotypes than others. However, the genetic background of these reactions still remains mostly unknown. One of the major aims of exercise genomics is to finally be able to define molecular markers, which by themselves or in combination with other biomarkers would make it possible to predict the benefits from an exercise program or a physically active lifestyle. Currently, some studies have tried to answer these questions, however, they still represent only the first steps towards an understanding of the gene variants x physical activity interactions. An understanding of the genetic determinants is the most promising tool for individualization of the training process for enhancing competitive athletic performance, sport selection, sport traumatology, and even illegal “gene doping”.
AMPD1 polymorphism was associated not only with obesity-related traits. AMPD deficiency connected with T allele also causes several exercise limitations. It was described as the reason for metabolic myopathy, early muscle fatigue, stiffness, myalgia, and delayed recovery of muscle strength. Consequently, the aim of the study was to determine if the C34T polymorphism influences the effects (not only on obesity-related traits) of an aerobic fitness program.
Previously, many authors described an association between this important genetic marker and athlete status. They suggested that the T allele is associated with a deficiency of AMPD and as a result, exercise limitation. However, there are a lack of articles concerning the potential impact of AMPD1 on the extent and nature of the response to training in healthy women. To the best of our knowledge, our research team analyzed this association for the first time. Therefore, the obtained results cannot be directly compared to other authors’ studies.
Skeletal muscle-specific AMPD, encoded by AMPD1 gene, is stimulated after abbreviated strenuous physical activity and is a key component of muscular energy metabolism while exercising, so this gene seems to be an important genetic marker of training-induced changes in the human body. We think that further studies are necessary to elucidate the role of the C34T polymorphism on the effectiveness of an exercise training program.
Second, it is not clear whether the diet is strictly controlled. The blood lipids and glucose are largely influenced by food intake. If the food is not strictly controlled, these data would be influenced by more than two factors (diet plus C34T and training).
Reply - We were not specific enough while describing the procedures used in the study. All the women had three mandatory meetings where they received specific guidance on rational nutrition and, if needed, individual tips. The participants kept a daily "diet diary" recording everything they drank and ate. Weekly consultations were held in which the quality and quantity of meals were analyzed and, if necessary, minor adjustments were made. It is worth mentioning that there were no overweight or obese women in the study group, therefore the diet mainly consisted of eating reasonable meals at specific times. Individuals started their diet and physical activity at the same time.
We added new information related to diet (L 101-105)
Third, the conclusion that ‘AMPD1 C34T polymorphism is not associated with obesity-related traits’ is not solid. According to the introduction, it is expected that T allele reduces obesity. If normal BMI people were recruited in this study, treatments (such as HFD) should be utilized to induce the obesity of these people, and then check the obesity-related parameters. The combination of normal people and aerobic fitness did not explain anything about obesity, for the room is limited for normal people to lose weight or fatty acids.
Reply -
Thank you for this comment. We fully agree with the reviewer, so we changed the conclusion: “The results did not confirm that harboring the rs17602729 T allele influences the effects of a training program.” (L 266-267)
Minors:
In Abstract, the second sentence ‘However… is unclear’ is confusing. Please revise.
Reply –change has been made
In the training program, whether the cycle speed is the same for every participant?
Reply - A continuous graded test on an electronic cycle ergometer was used only before the training program to assess the maximum heart rate.
Consider to do a the 3 × 2 [genotype (CC, CT and TT) × training interactions] analysis.
Reply – The 3 × 2 [genotype (CC, CT and TT) × training interactions] analysis was not included in the analysis owing to a low number of rare homozygotes, only 4% of the participants were TT homozygotes (this is in accordance with other studies which show that ~2% of the general Caucasian population are TT homozygotes).
Round 2
Reviewer 2 Report
The manuscript has been greatly improved after revision. I agree to publish it in the current version.